# Cow’s Milk Intake and Risk of Coronary Heart Disease in Korean Postmenopausal Women

**DOI:** 10.3390/nu14051092

**Published:** 2022-03-05

**Authors:** Ae-Wha Ha, Woo-Kyoung Kim, Sun-Hyo Kim

**Affiliations:** 1Department of Food Science and Nutrition, College of Natural Science, Dankook University, Cheonan City 31116, Chungcheongnam-do, Korea; aewhaha@dankook.ac.kr (A.-W.H.); wkkim@dankook.ac.kr (W.-K.K.); 2Department of Technology and Home Economics Education, Kongju National University, Gongju City 32588, Chungcheongnam-do, Korea

**Keywords:** milk, coronary heart disease (CHD), calcium, FRS, atherogenic index, HDL cholesterol

## Abstract

Numerous studies have reported conflicting results associated with cow’s milk intake and coronary heart disease (CHD). However, studies involving postmenopausal women are very limited. This study was therefore undertaken to identify the relationship between cow’s milk intake and CHD risk in postmenopausal women, using data from the 6th period of the Korea National Health and Nutrition Examination Survey (2013–2015). A total of 1825 postmenopausal women, aged 50–64 years old, were included in the final analysis. The frequency of cow’s milk consumption for each subject was determined using the semi-quantitative food frequency questionnaire, and was classified into four groups (Q1–Q4): Q1, group that did not drink milk (no milk, n = 666); Q2, 0 < frequency of milk intake per week ≤ 1 (n = 453); Q3, 1 < frequency of milk intake per week ≤ 3 (n = 319); and Q4, frequency of milk intake >3 times per week (n = 387). General characteristics, such as education, living area, household income, and obesity level, were compared between the four groups. Percentages of daily nutrient intake compared to the dietary reference intake for Koreans (KDRIs) were determined, and the Framingham Risk Score (FRS), atherogenic index (AI), and atherogenic index of plasma (AIP) were determined as the CHD risk indicators. Except household income, no significant difference was obtained among the four groups with respect to age, education, living area, or obesity. Compared to KDRIs, the intake ratio of calcium, phosphorus, and riboflavin were significantly higher in the Q4 group than in the Q1–Q3 groups. Blood HDL-cholesterol was significantly higher in Q4 than in Q1. The CHD risk factors FRS (%), AI, and AIP were significantly lower in the Q4 group as compared to the other groups (CHD risk (%): Q1 9.4, Q4 8.5; AI: Q1 3.06, Q4 2.83; API: Q1 0.37, Q2 0.31, Q4 0.32). FRS was determined to be significantly and positively correlated to AI or AIP, and negatively correlated with the cow’s milk intake frequency and calcium intake. In conclusion, compared to women who do not consume cow’s milk, postmenopausal women who consume cow’s milk frequently have a better nutritional status of calcium, phosphorus, and vitamin B12, higher HDL levels, and a lower level of CHD risk indicators, such as FRS, AI, and AIP, contributing to decreased CHD risk in a 10-year period. Therefore, to prevent the risk of CHD in postmenopausal women, there needs to be a greater emphasis for cow’s milk consumption four or more times per week.

## 1. Introduction

Menopause is an important event, wherein women experience emotional and physiological changes due to decreased estrogen secretion [1]. Moreover, there is a significantly increased risk of coronary heart disease (CHD) with menopause [2,3,4,5,6]. Premenopausal women have a significantly lower incidence of CHD than men of the same age. However, from the age of 50 years onwards (the age at which menopause generally begins), this risk increases to a similar level as seen in men, with a higher incidence of CHD being observed in women above 75 years [2,3]. The increased risk of CHD is related to the altered blood lipid profiles observed in postmenopausal women [4]. At menopause, the basal metabolic rate is also lowered, which increases the risk of obesity and enhances the risk of CHD [5]. 

The traditional risk factors for CHD include dyslipidemia, hypertension, diabetes, obesity, family history, and smoking, most of which can be controlled through individual lifestyle changes [3]. To prevent CHD, it is recommended to increase the intake of vegetables, fruit, and complex carbohydrates, and reduce the intake of foods containing saturated fat and cholesterol [6]. Although milk is a nutrient-dense food, there have been conflicting results on the relationship between milk/dairy product intake and CHD [7,8,9,10,11,12,13,14]. Some studies have reported that the excessive consumption of milk and dairy products increases the risk of CHD because of the high saturated fat content [7,8,9]. Positive effects on milk and dairy products and CHD have also been reported; sufficient intake of milk and dairy products was found to lower blood pressure and increase HDL-cholesterol levels [10,11]. The effects of milk and dairy products on the risk of CHD differ according to the type of dairy product consumed (milk, yogurt, and cheese), or milk fat content (low-fat milk /regular milk) [12,13,14].

The index of dietary reference intake for Koreans (KDRIs, Ministry of Health and Welfare, The Korean Nutrition Society, Dietary reference intakes for Koreans 2020, Sejong; 2020.) is an energy and nutrient intake standard set for maintaining and promoting the health of Korean people. To evaluate the adequacy of nutritional status, the subject’s daily nutrient intake is generally calculated as % KDRIs. According to the 2020 KDRIs, the recommended intake of calcium for women aged 50–64 years is 800 mg/day; one cup (200 mL) of cow’s milk contains approximately 200 mg, which is 28.6% of the daily recommended intake of calcium. Calcium is the most deficient nutrient for Koreans [15]. Therefore, the Korean Nutrition Society recommends drinking one cup (200 mL) of cow’s milk (or l servings of cheese (40 g) or liquid (150 mL)- /solid (100 g)-yogurt) per day for adults, and two cups (400 mL) or more for growing children or pregnant women. The recommended amount is 100 mg higher for postmenopausal women than for adult women (700 mg/day). Hence, it is recommended to consume more than 1 cup of cow’s milk.

The Framingham Risk Score (FRS) is a tool that predicts the risk of CHD for a 10-year period, considering various factors such as gender, age, smoking status, diabetes status, blood pressure, total blood cholesterol, and HDL-cholesterol [16,17]. Since CHD-risk prediction through FRS is calculated by considering multiple factors rather than a single factor, it is possible to predict the risk accurately and efficiently. The validity of FRS has been verified in previous studies [16,17,18,19,20]. To date, most studies on milk and dairy products and CHD have been conducted on adults, and studies on milk intake and CHD in postmenopausal women are very limited. Therefore, the current study was undertaken to determine the relationship between milk intake and nutritional status, compared with %KDRI, and CHD risk, determined as FRS, AI, and AIP, in postmenopausal women using data from the 2013–2015 Korea National Health and Nutrition Examination Survey (KNHANES). 

## 2. Materials and Methods

### 2.1. Data Collection and Selection of Study Subjects

Data from the 6th period (2013–2015) of KNHANES was used for this study. The KNHANES is a national surveillance system that has been assessing the health and nutritional status of Koreans since 1998. Since then, the frequency of redesigning KNHANES has increased from once every 3 years to every year. Data for this study were downloaded from the KNHANES website (http://knhanes.cdc.go.kr, accessed on: 30 June 2020), and this was conducted with the approval of the Research Ethics Review Committee (KNU_IRB_2020-64, Konju National University, Korea).

Of the total 22,948 subjects included (2013–2015 KNHANES), males (n = 10,411), missing data, and indeterminate answers (answers marked as 9 (Don’t know, no answer) to certain questions) were excluded from the study. Among the remaining women (n = 12,294), only subjects aged 50–64 years who answered the question “Are you menstruating now?” as “natural menopause and entered the age of menopause” were selected for this study (n = 2016). Of these, subjects who were unable to provide data on milk consumption frequency were excluded (n = 191, including extreme energy intakes (<500 Kcal or >5000 Kcal)). Ultimately, data obtained from 1825 women were included in the final analysis. 

### 2.2. Analysis of Milk Intake Frequency

The nutrition survey in KNHANES uses the face-to-face interview method and includes dietary behaviors, food frequency, and food intake. The semi-quantitative food frequency questionnaire (FFQ) comprises 63 food items that are key sources of energy and nutrients. The semi-quantitative FFQ is designed as an open-ended survey for reporting various dishes and foods using the 24-h recall method with various measuring aids. In this study, the cow’s milk consumption frequency of the subjects was determined using the semi-quantitative FFQ. The frequency of cow’s milk intake (including low-fat milk) was divided into nine frequencies, as follows: rarely, once a month, 2–3 times a month, once a week, 2–4 times a week, 5–6 times a week, once a day, twice a day, and three times a day. The data were converted into weekly cow’s milk intake frequency and daily cow’s milk intake (mL/d). 

### 2.3. General Characteristics of the Subjects According to Cow’s Milk Intake Frequency

Based on the frequency of cow’s milk intake per week, total subjects were classified into four groups: Q1, group that did not drink cow’s milk (no milk, n = 666); Q2, 0 < frequency of milk intake per week ≤ 1 (n = 453); Q3, 1 < frequency of milk intake per week ≤ 3 (n = 319); and Q4, frequency of milk intake > 3 times per week (n = 387). General characteristics, including education level, living area, household income, and obesity level, were compared among the groups. The education level was divided into four groups: elementary school graduation or lower, middle school graduation, high school graduation, and college graduation or higher. The subjects were classified into low, middle–low, middle–high, and high as the household income quartile according to the sample household and the income quartile standard amount of the sampled population in the KNHANES. The living area was divided into three parts: large cities, small and medium-sized cities, and rural areas. Obesity was classified into three categories according to body mass index (BMI, weight (kg)/ height (m)^2^) values: less than 18.5 kg/m^2^ (underweight), less than 18.5–24.9 kg/m^2^ (normal weight), and over 25 kg/m^2^ (obesity).

### 2.4. Evaluation of Nutrient Intakes Compared with the KDRIs 

Daily energy and nutrient intake were determined, and the ratio of energy to carbohydrate, protein, and fat intake was also calculated. The KDRIs included four levels: estimated average requirement (EAR), recommended nutrient intake (RNI), adequate intake (AI), and tolerable upper intake level (UL). To evaluate their adequacy, the energy and nutrient intakes for 11 nutrients (energy, protein, calcium, phosphorus, iron, sodium, vitamin A, thiamine, riboflavin, niacin, and vitamin C) were compared with the age- and gender-specific values of the KDRIs. For energy, the intake ratio to the estimated energy requirement (EER) was calculated. For nutrients, the intake ratio to the RNI or AI of each nutrient was calculated.

### 2.5. Anthropometric Measurements and Blood Profiles

Height, weight, body mass index (BMI), and blood pressure were analyzed according to the frequency of cow’s milk intake. The BMI was calculated as body weight in kilograms divided by the square of the body height in meters (kg/m^2^). Levels of fasting blood glucose, glycosylated hemoglobin, total cholesterol, triglyceride, LDL-cholesterol, and HDL-cholesterol were analyzed from the hematological data obtained. 

### 2.6. Smoking Status and Disease Fraction (%)

The smoking status was classified as yes or no to the survey question of “Do you currently smoke?”. The presence or absence of disease (diabetes, hypertension, and hyperlipidemia) was classified as ‘yes’ for each disease diagnosed by the doctor and ‘no’ if absent.

### 2.7. Framingham Risk Score (FRS), Atherogenic Index (AI), and Atherogenic Index of Plasma (AIP)

Factors such as age, smoking status, diabetic status, blood pressure, total blood cholesterol, and blood HDL-cholesterol were included for calculating the FRS total score. The range of scores for age was −9 to 8, and scores were given by dividing the 30–74-year-olds into five-year increments. Blood HDL-cholesterol (score range: −3 to 5 points) was classified as <35 mg/dL, 35–44 mg/dL, 45–49 mg/dL, 50–59 mg/dL, and ≥60 mg/dL. Total blood cholesterol was categorized into the following five levels (<160 mg/dL, 160–199 mg/dL, 200–239 mg/dL, 240–279 mg/dL, and ≥280 mg/dL) and scored in the range −2 to 3 points. Smokers were scored as per their smoking status (smokers, non-smokers; range: 0 to 2 points). Blood pressure and systolic blood pressure were divided into the following 5 stages: <120 mmHg, 120–129 mmHg, 130–139 mmHg, 140–159 mmHg, and ≥160 mmHg, and scores were assigned regarding whether hypertension was treated or untreated (−1 to 7 points). Diabetes was also scored by considering the presence or absence of diabetes (score range: 0–4 points). The 10-year CHD risk (%) was calculated according to the sum of the FRS points for the above five indicators. The atherogenic index (AI) and atherogenic index of plasma (AIP) were also analyzed to evaluate the CHD risk and were calculated using the following formulae: AI = (non-blood HDL-cholesterol)/blood HDL-cholesterol. AIP was calculated as the logarithm of the ratio of blood triglyceride (mg/dL) to blood HDL-cholesterol (mg/dL).

### 2.8. Statistical Analysis

Data were analyzed by considering the integrated weight of KNHANES (stratified multistage probability sampling), strata (KSTRATA), and cluster (primary sampling unit, PSU). Continuous variables are expressed as the mean and standard error, and significance was verified by proc survey regression after adjusting for age, household income, and energy intake. The differences between groups were verified by performing Bonferroni’s post-hoc test at the level of α = 0.05. Categorical variables are expressed as frequency and percentage, and significance was verified by the χ^2^-test. Correlation between cow’s milk intake frequency and variables was analyzed using the Pearson’s partial correlation (control: age, energy intake, household income). All statistical analyses were performed using the SAS version 9.4 (Statistical Analysis System, SAS Institute, Cary, NC, USA). 

## 3. Results

### 3.1. General Characteristics of Subjects According to Cow’s Milk Intake Frequency 

Subjects were divided into four groups based on the frequency of cow’s milk consumption per week: Q1, no milk (n = 666); Q2, 0 < weekly frequency ≤ 1 (n = 453); Q3, 1 < weekly frequency ≤ 3 (n = 319); and Q4, weekly frequency > 3 (n = 387) (Table 1). Daily cow’s milk intake and frequency of cow’s milk intake in each group were determined to be 0 g and 0 times per week in Q1, 18.2 g and 0.7 times per week in Q2, 85.9 g, and 3.0 times per week in Q3, and 215.9 g and 7.2 times per week in Q4, respectively. No significant difference was obtained for mean age among the groups. Except household income, no significant difference was determined for age, education, living area, or obesity when comparing the groups. Higher household income entailed a greater number of subjects belonging to the Q4 group (*p* = 0.0012). 

### 3.2. Nutrient Intakes Compared with the KDRIs According to Cow’s Milk Intake Frequency

Nutrient intakes of the subjects were compared with the corresponding KDRIs considering gender and age. All subjects were consuming more than the recommended KDRIs for all nutrients, except calcium and riboflavin (Table 2). The intakes of calcium, phosphorus, and riboflavin were significantly higher in the Q4 group, as compared to the Q1 group. The % KDRIs of calcium were determined to be the lowest in the Q1 group, and highest in the Q4 group. Overall, the calcium % KDRIs of total subjects was 59.8%. 

### 3.3. Anthropometric Measurements and Blood Profiles According to the Cow’s Milk Intake Frequency 

No significant differences in height, weight, or BMI were observed among the four groups (Table 3). Moreover, there were no significant differences between the four groups in values determined for the indicators of diabetes (fasting blood glucose level and glycated hemoglobin). Evaluation of the blood lipid profile revealed a significant difference only for HDL-cholesterol levels among the groups; the blood HDL-cholesterol concentration was significantly higher in the Q4 group as compared to the Q1 group. Blood triglyceride concentrations were lower in the Q2~Q4 group than in the Q1 group, but failed to reach statistical significance. No significant difference was obtained among the groups for systolic and diastolic blood pressure.

### 3.4. Smoking Status and Disease Fraction (%) According to Cow’s Milk Intake Frequency

Current smokers comprised 3.3% of the total subjects, without any statistical difference among groups (Table 4). The prevalence of diabetes was 8.2%, hypertension 26.1%, and hyperlipidemia 13.8%. There was no significant difference in the prevalence of diabetes, hypertension, and hyperlipidemia between the four groups of cow’s milk intake frequency. 

### 3.5. Indicators for CHD Risk According to Cow’s Milk Intake Frequency

Among the FRS indicators, no significant differences were obtained between groups with respect to age, total cholesterol, blood pressure, diabetes mellitus, or smoking (Table 5). However, the blood HDL-cholesterol score showed statistical significance according to the frequency of cow’s milk intake: the blood HDL-cholesterol score in the Q4 group (0.4 ± 0.1) was significantly lower than that of the Q1 group (0.1 ± 0.1). The average FRS in all study subjects was 8.6±0.1. The FRS score of the Q4 group was 8.3 ± 0.2, which was significantly lower than the Q1 group (FRS: 8.9 ± 0.2). In the 10-year CHD risk (%) calculated according to the total FRS score, the Q4 group had significantly lower CHD risk (%) than the Q1 group (Q1 9.4 ± 0.3, Q4 8.5 ± 0.2, average 8.9 ± 0.2). The mean AI and AIP were 2.95 ± 0.03 and 0.33 ± 0.01, respectively, and both values of the Q4 group were significantly high compared to values of the Q1 group (AI: Q1 3.06 ± 0.04 vs. Q4 2.83 ± 0.06; API: Q1 0.37 ± 0.01 vs. Q4 0.32 ± 0.02).

### 3.6. Correlations between Cow’s Milk Intake, Calcium Intake, AI, FRS, and AIP

After adjusting for energy, age, and household income, the correlations between cow’s milk frequency (times/week), calcium intake (g/d), FRS, AI, and AIP were analyzed by Pearson correlation (Table 6). The frequencies of cow’s milk intake and calcium intake were significantly and negatively correlated with the CHD risk indicators, such as FRS and AI. Furthermore, a significant positive correlation was determined among the CHD indicators FRS, AI, and AIP. 

## 4. Discussion

Coronary heart disease (CHD) is a disease of the cardiovascular system, also referred to as ischemic heart disease (IHD) and coronary artery disease (CAD). CHD, such as myocardial infarction and angina pectoris, accounts for 60% of deaths from heart disease [2]. The prevalence and the mortality of CHD in elderly women are increasing continuously [2,3,4]. CHD is caused by insufficient blood supply to a part of the heart muscle caused by narrowing the coronary arteries, causing atherosclerosis of the coronary arteries [2]. Therefore, it is important to improve the eating habits that cause arteriosclerosis to prevent CHD and slow the progression of CHD. Despite the many studies on milk and CHD, the research results are contradictory, and studies on postmenopausal women are limited [7,8,9,10,11,12,13,14]. Thus, this study determined the relationship between cow’s milk intake and CHD risk in postmenopausal women using the KNHANES data from 2013–2015, when FFQ data were provided for that period.

Dietary survey methods include the 24-h recall method and the FFQ method [21] The 24-h recall method has a drawback in that the investigation is limited to one day. Semi-quantitative FFQ allows an investigation of general eating habits through a single survey [21]. Studies have reported that large-scale studies based on the FFQ might underestimate energy or some nutrients because FFQ utilizes a fixed food list [21,22]. In this study, pproximately 80% of postmenopausal women did not consume milk on the day of the study using 24-h recall data. Only 36% of the subjects in FFQ data analysis did not drink milk. Previous Korean studies [23,24] also reported similar results to this study. Therefore, in this study, subjects were recruited using the weekly cow’s milk intake frequency of FFQ, but the energy and nutrient intakes were determined from the 24-h recall data of the subjects. 

In this study, the Q4 group with the highest cow’s milk intake had higher intakes of calcium, phosphorus, and riboflavin nutrients (Table 2), as well as higher blood HDL cholesterol concentration (Table 3), as compared to the Q1 group who did not consume cow’s milk. Moreover, we obtained significant positive correlations between cow’s milk intake frequency and calcium intake (mg/d) (Table 6), thereby confirming that cow’s milk intake is a major source of calcium for Korean postmenopausal women. 

Limited studies have previously reported that milk and dairy products or calcium intake are related to improving blood lipid levels in postmenopausal women [25,26,27,28]; one clinical study found that the consumption of cow’s milk (two cups per day) in postmenopausal women lowered the CHD risk by improving blood lipid profiles [25]. Daily intakes of calcium or dairy products also helped in reducing the risk of metabolic syndrome in postmenopausal women by improving the blood HDL-cholesterol and decreasing blood cholesterol (TC) and triglyceride (TG) [26,27,28]. The effect of calcium or dairy products on the lipid profile remains unclear, but it has been suggested that increased calcium intake from cow’s milk results in inhibiting fat absorption and increasing fecal fat excretion, which is associated with reduced blood LDL-cholesterol and increased HDL-cholesterol [29]. However, it is necessary to clarify this correlation through prospective and experimental studies in the future. 

The association between altered blood lipid profiles and CHD has been well established through previous studies [2,3,4,5,6,15,16,17,24,25,26,27,28,29]. Thus, blood lipids, such as TC, LDL-cholesterol, HDL-cholesterol, and TG, are widely used as indicators for predicting CHD risk [15]. Atherogenic index (AI) predicts CHD risk using blood TC and LDL-cholesterol [30], whereas the atherogenic index of plasma (AIP) calculates the risk with TG and HDL-cholesterol, which have recently been identified as strong indicators of CHD [31]. Increased TG levels in blood cause an increase in small dense LDL levels and an ultimate increase in CHD risk [15]. AIP < 0.11 or AI < 3 are reported to indicate low CHD risk [15,30,31,32].

The Framingham Risk Score (FRS) predicts CHD risk for the next 10 years by applying the blood lipid components as well as other factors such as diabetes, smoking, gender, age, and blood pressure [15,16,17,18], classifying the CHD risk as low (FRS < 10%), moderate (10% < FRS < 20%), or high (FRS) > 20%) [17]. For calculating the FRS total score, either TC or LDL-cholesterol are selected. Thus, the total FRS score varies, depending on whether LDL-cholesterol or TC are applied. It has been confirmed that FRS is an effective tool for predicting CHD in Korean adults [6,18,20,33,34,35,36,37], and it was suggested that calculating FRS using TC is more suitable than using LDL-cholesterol, especially in women [33]. Therefore, in the current study, FRS was calculated using TC and the FRS (%), as CHD risk in Korean menopausal women was 8.9% (Table 6) and 64.7% belonged to the low CHD risk (data are not shown).

The FRS (%) in foreign menopausal women has been reported as 12–15% [9,15,16,17,18,30,31,32], thereby indicating that the CHD risk in Korean menopausal women is low (8.9), as compared to foreign women. The difference in FRS (%) between this and other studies may be due to differences in the race and age of the study subjects. Age is an important factor in FRS calculations. In a small number of postmenopausal women in the United States, the FRS (%) was 15.1 and the mean age of the subjects was 60 years. In that study, 30% of subjects were classified in the low-risk group for CHD [19]. In another study of postmenopausal women in Cameroon, subjects belonged to the 45–80 years of age group, with a mean age of 56.4 years. The FRS (%) was determined to be 13.4, and 39.8% of subjects had a low risk of CHD [30]. 

Previous studies have reported that FRS (%) could underestimate CHD risk in women [30,31,32,33]. Therefore, in our study, to determine the validity of FRS (%) as a predictive risk indicator for CHD, other indicators such as AI and AIP were also measured. Our results revealed a significant correlation between FRS and AI (r = 0.575, *p* < 0.001) as well as FRS and AIP (r = 0.540, *p* < 0.001) (Table 6). This validates the appropriateness of FRS as an indicator for predicting the risk of CHD in Korean postmenopausal women aged 50–64 years. Moreover, we also found that CHD risk indicators (FRS and AI) in the group who did not drink cow’s milk were significantly higher than the Q4 group (highest cow’s milk intake group), indicating that increased cow’s milk intake was related to a reduction in CHD risk.

Types of cow’s milk (non-fat/low fat/whole milk) and types of dairy product (milk, yogurt, cheese) consumed in adults, not postmenopausal women, are associated differentially with CHD [12,13,14]. Because different types of dairy products (milk, cheese, and yogurt) have different nutritional content, recent studies have focused on the effects of CHD from specific types of dairy products rather than total dairy intake [12,13,14]. Some studies reported a positive association of high-fat milk and CHD risk and an inverse association of low-fat milk/non-fat with CHD risk [7,8,9]. Other studies showed that a higher milk intake, regardless of the fat content of whole milk, was associated with a lower risk of CHD [10,11,12,38], or had a neutral association [13,14,39]. Publication bias due to the characteristics of meta-analysis studies could have underestimated or overestimated the CHD risks associated with milk consumption [39].

In this study, among the milk drinkers, the proportion of regular milk (45.7%), low-fat milk (27.4%), and both (10.1%) were determined (data not shown), and there were no significant differences in the ratios in the milk type between the milk groups (Q2~Q4). In addition, there were insufficient numbers of postmenopausal women to study CHD risk by milk type for this study. Thus, in this study, the milk intake group was not divided into low-fat milk and whole milk, and instead, the milk groups that consumed both low-fat and regular milk were included, which is a limitation of this study. 

In the case of Korea, the risk of CHD caused by the excessive consumption of milk (low-fat milk or whole milk) is considered lower than that of Western countries because milk intake is still insufficient in all ages [24,25,28]. Nevertheless, studies still suggest that the excessive consumption of high-fat milk may increase the risk of CHD. Therefore, low-fat milk should be consumed by people with a genetic factor for CHD [33]. A future study will examine the type of milk (low-fat milk, whole milk, or both) intake and CHD risk. 

This study made the following efforts to minimize errors in the statistical results. The stepwise regression was conducted to examine the statistical significance of other independent variables (as confounding factors, such as age, energy intake, household income, BMI, living area, and education level) in a linear regression model (dependent variable: CHD risk; independent variable: milk intake frequency). As a result, three variables—age, energy intake, and household income—were significant. Accordingly, multiple linear regression was performed after adjusting for these confounding factors. Multicollinearity between these variables was proven by collinearity statistics (tolerance, variance inflation factors (VIF)). Each VIF of those variables was between 1–2, suggesting non-multicollinearity. Gender and age are the major variables that affect FRS score. Therefore, the study subjects were limited to only women aged 50–64 years old. Even within the age range, data were analyzed statistically after adjusting for age to minimize statistical errors. In addition, diseases (diabetes, hyperlipidemia, and hypertension) are major factors of CHD. As listed in Table 4, there were no differences in disease fractions between the four groups, suggesting that the low CHD risk (%) in the Q4 group was more likely to be related to the high milk intake than other factors such as disease. 

Nevertheless, since this was a cross-sectional study, the following limitations must be noted. First, a causal relationship between cow’s milk intake and CHD risk could not be determined. Second, since this was a cross-sectional study, we were unable to evaluate the accuracy of FRS in predicting CHD events. Self-reported FFQ, 24-h recall, and status of menopause were also limitations of this study. In the FFQ used in this study, only milk (including low-fat milk and regular milk)/number of intakes was assessed. Therefore, it is unclear if the milk consumed by the study subjects was low-fat or regular milk. Diet (high-fat foods, fruits, vegetables, and whole grains) could be an influencing factor for CHD risk, but this was not considered in this study. However, to overcome the limitations, this study used large-scale national data (KNHENES), variables that could affect statistics were controlled to the maximum, and not just FRS but various other CHD risk indicators such as AI and AIP were analyzed.

## 5. Conclusions

The Korean postmenopausal women (50–64 years old) who consumed cow’s milk four times or more per week had higher calcium, phosphorus, and riboflavin intakes than those who did not consume milk. In addition, they had significantly higher HDL-cholesterol levels in the blood. Moreover, the levels of the CHD risk indicators, such as FRS and AI, were significantly lower in the high cow’s milk intake group. These findings highlight the need for further studies to assess cow’s milk as an intervention for postmenopausal women to reduce CHD risk.

## Figures and Tables

**Table 1 nutrients-14-01092-t001:** The general characteristics of the subjects according to cow’s milk intake frequency.

Variables		Q1 ^1^ (n = 666)	Q2 (n = 453)	Q3 (n = 319)	Q4 (n = 387)	Total (n = 1825)	*p* Value ^2^
Daily cow’s milk intake (mL/d) (mean ± SE)		0.0 ± 0.0 ^3a4^	18.2 ± 0.5 ^b^	85.9 ± 1.2 ^c^	215.9 ± 4.6 ^d^	66.8 ± 2.4	<0.0001 ***
Age (year) (mean ± SE)		57.1 ± 0.2	56.8 ± 0.3	56.3 ± 0.3	57.1 ± 0.3	57.0 ± 0.1	0.1453
Education Level (n (%))	≤Elementary school	273(36.8) ^5^	130(27.2)	85(25.0)	109(26.4)	597(30.1)	0.6229
	Middle school	129(21.1)	109(24.3)	79(24.2)	94(23.7)	411(23.0)	
	High school	181(30.3)	157(35.3)	109(36.9)	119(31.3)	566(32.9)	
	≥College	83(11.9)	57(13.2)	46(13.9)	65(18.5)	251(14.0)	
	Total	666(100.0)	453(100.0)	319(100.0)	387(100.0)	1825(100.0)	
Household income (n (%))	Low	146(20.0)	66(13.8)	45(13.7)	44(11.4)	301(15.6)	0.0012 **
	Middle–low	181(28.7)	131(28.4)	82(23.6)	110(25.1)	504(26.9)	
	Middle–high	165(24.5)	127(27.8)	88(28.7)	99(26.0)	479(26.4)	
	High	172(26.8)	129(30.0)	101(34.0)	133(37.5)	535(31.1)	
	Total	664(100.0)	453(100.0)	316(100.0)	386(100.0)	1819(100.0)	
Region (Living area) (n (%))	Large city	303(46.6)	223(49.7)	164(53.3)	172(43.9)	862(48.0)	0.0928
	Middle and Small city	218(33.5)	136(30.7)	108(34.3)	147(39.2)	609(34.2)	
	Rural area	145(19.9)	94(19.6)	47(12.4)	68(16.9)	354(17.8)	
	Total	666(100.0)	453(100.0)	319(100.0)	387(100.0)	1825(100.0)	
Obesity ^6^ (n (%))	Underweight	20(3.5)	4(1.2)	6(1.9)	5(1.5)	35(2.2)	0.0570
	Normal weight	415(64.4)	280(64.0)	204(63.8)	245(63.0)	1144(63.9)	
	Obese	230(32.2)	168(34.8)	109(34.3)	137(35.5)	644(33.9)	
	Total	665(100.0)	452(100.0)	319(100.0)	387(100.0)	1823(100.0)	

^1^ Q1: No milk intake, Q2: 0 < times/week ≤1, Q3: 1 < times/week ≤ 3, Q4: 3 > times/week. ^2^
*p* value was determined with proc survey multiple regression for the continuous variables, while *p* value by chi-square test was used for the categorized variables (** *p* < 0.01 *** *p* < 0.001). ^3^ Mean ± SE. ^4^ abcd: values with different letters in the same row are significantly different at *p* = 0.05 by Bonferroni test. ^5^ N (%). ^6^ Obesity, divided by body mass index (BMI, kg/m^2^): underweight, BMI < 18.5; normal weight, 18.5 ≤ BMI ≤ 24.9, obese, BMI ≥ 25.0.

**Table 2 nutrients-14-01092-t002:** The % KDRIs of nutrient intake according to cow’s milk intake frequency.

Variables ^3^	Q1 ^1^ (n = 666)	Q2 (n = 453)	Q3 (n = 319)	Q4 (n = 387)	Total (n = 1825)	*p* Value ^2^
Energy	89.8 ± 1.5 ^4^	99.2 ± 2.0	99.0 ± 2.1	103.5 ± 2.0	97.0 ± 0.9	0.8562
Protein	119.4 ± 2.5	134.0 ± 3.3	134.7 ± 3.5	146.3 ± 4.3	132.2 ± 1.7	0.0802
Calcium	50.0 ± 1.2 ^a5^	56.7 ± 1.6 ^a^	62.1 ± 1.8 ^b^	76.0 ± 2.1 ^c^	59.8 ± 0.9	<0.0001 ***
Phosphorus	128.6 ± 2.5 ^a^	140.8 ± 3.1 ^a^	147.3 ± 3.6 ^b^	163.9 ± 4.0 ^c^	143.0 ± 1.6	<0.0001 ***
Iron	194.3 ± 4.4	206.1 ± 5.8	203.7 ± 5.5	209.4 ± 5.7	202.1 ± 2.7	0.3764
Sodium	203.8 ± 5.6	234.3 ± 10.8	218.3 ± 7.9	233.2 ± 9.2	220.8 ± 4.2	0.5958
Vitamin A	108.6 ± 5.2	118.9 ± 6.6	123.3 ± 7.7	153.7 ± 20.6	127.2 ± 5.3	0.4879
Thiamin	145.0 ± 3.1	156.8 ± 3.9	156.7 ± 4.2	167.2 ± 4.2	154.8 ± 1.9	0.7107
Riboflavin	70.5 ± 1.7 ^a^	79.5 ± 2.1 ^a^	83.1 ± 2.6 ^b^	97.3 ± 2.6 ^c^	80.6 ± 1.2	<0.0001 ***
Niacin	96.9 ± 2.2	107.9 ± 2.8	109.2 ± 3.3	111.5 ± 3.1	105.0 ± 1.4	0.8911
Vitamin C	131.0 ± 6.4	132.6 ± 8.4	135.8 ± 9.3	156.7 ± 11.0	137.8 ± 4.6	0.4170
Energy from carbohydrates (%)	71.0 ± 0.5 ^a^	68.8 ± 0.6 ^b^	67.7 ± 0.7 ^b^	67.4 ± 0.6 ^b^	69.1 ± 0.3	0.0002 **
Energy from protein (%)	13.5 ± 0.2	13.8 ± 0.2	13.8 ± 0.2	14.3 ± 0.2	13.8 ± 0.1	0.1820
Energy from fat (%)	15.5 ± 0.3 ^a^	17.4 ± 0.5 ^b^	18.5 ± 0.6 ^b^	18.3 ± 0.5 ^b^	17.1 ± 0.2	0.0001 **

^1^ Q1: No milk intake, Q2: 0 < times/week ≤ 1, Q3: 1 < times/week ≤ 3, Q4: 3 > times/week. ^2^
*p* value by proc survey regression after adjusting for age, energy intake, and household income (** *p* < 0.01 *** *p* < 0.001). ^3^ Dietary reference intakes for Korean (KDRIs): energy, estimated energy requirement (EER); protein, Ca, P, Fe, vitamin A, thiamin, riboflavin, niacin, vitamin C, recommended nutrient intake (RNI); sodium, adequate intake (AI). ^4^ Mean ± SE. ^5^ abc: values with different letters in the same row are significantly different at *p* = 0.05 by Bonferroni test.

**Table 3 nutrients-14-01092-t003:** Anthropometric measurements and blood profiles according to cow’s milk intake frequency.

Variables	Q1 ^1^ (n = 666)	Q2 (n = 453)	Q3 (n = 319)	Q4 (n = 387)	Total (n = 1825)	*p* Value ^2^
Height (cm)	155.5 ± 0.2 ^3^	156.2 ± 0.3	156.1 ± 0.3	156.0 ± 0.3	155.9 ± 0.1	0.4929
Weight (kg)	57.9 ± 0.4	58.6 ± 0.4	58.7 ± 0.5	57.9 ± 0.5	58.2 ± 0.2	0.3544
Body Mass index (kg/m^2^)	24.0 ± 0.2	24.1 ± 0.2	24.2 ± 0.2	23.93 ± 0.2	24.0 ± 0.1	0.5507
Fasting blood glucose (mg/dL)	99.1 ± 0.8	101.2 ± 1.2	102.0 ± 1.7	101.4 ± 1.4	100.9 ± 0.6	0.1396
Hemoglobin A1c (%)	5.9 ± 0.0	5.9 ± 0.0	5.9 ± 0.0	5.9 ± 0.0	5.9 ± 0.0	0.5511
Total cholesterol (mg/dL)	201.1 ± 1.5	200.5 ± 1.7	200.4 ± 2.2	202.4 ± 2.0	201.0 ± 0.9	0.8819
Triglyceride (mg/dL)	137.2 ± 3.8	122.2 ± 4.0	125.1 ± 5.6	125.7 ± 4.0	129.1 ± 2.0	0.0586
LDL-cholesterol (mg/dL)	123.3 ± 1.9	122.1 ± 2.5	124.5 ± 3.1	122.2 ± 2.8	122.8 ± 1.2	0.8818
HDL-cholesterol (mg/dL)	51.5 ± 0.5 ^a4^	53.3 ± 0.7 ^ab^	53.4 ± 0.8 ^ab^	55.5 ± 0.7 ^b^	53.2 ± 0.3	0.0002 **
Systolic blood pressure (mmHg)	120.5 ± 0.7	118.5 ± 0.8	118.9 ± 0.9	119.1 ± 0.9	119.6 ± 0.4	0.5947
Diastolic blood pressure (mmHg)	76.4 ± 0.4	75.6 ± 0.5	75.7 ± 0.6	75.7 ± 0.5	76.0 ± 0.3	0.6264

^1^ Q1: No milk intake, Q2: 0 < times/week ≤ 1, Q3: 1 < times/week ≤ 3, Q4: 3 > times/week. ^2^
*p* value by proc survey multiple regression adjusted for age, energy intake, and household income (** *p* < 0.01). ^3^ Mean ± SE. ^4^ ab: values with different letters in the same row are significantly different at *p* = 0.05 by Bonferroni test.

**Table 4 nutrients-14-01092-t004:** Smoking status and disease fraction (%) according to cow’s milk consumption frequency (N (%)).

Variables		Q1 ^1^ (n = 666)	Q2 (n = 453)	Q3 (n = 319)	Q4 (n = 387)	Total (n = 1825)	*p* Value ^2^
Smoking status	No ^3^	643(96.8)	436(97.0)	312(97.3)	372(95.6)	1763(96.7)	0.4123
	Yes	20(3.2)	16(3.0)	6(2.7)	13(4.4)	55(3.3)	
	Total	663(100.0)	452(100.0)	318(100.0)	385(100.0)	1818(100.0)	
Diabetes	No ^4^	613(93.2)	408(90.5)	294(90.6)	353(91.8)	1667(91.8)	0.4492
	Yes	53(6.7)	45(9.5)	25(9.4)	34(8.2)	157(8.2)	
	Total	666(100.0)	453(100.0)	319(100.0)	387(100.0)	1825(100.0)	
Hypertension	No ^4^	476(71.9)	341(75.3)	238(75.8)	282(74.2)	1337(73.9)	0.5687
	Yes	190(28.1)	112(24.7)	81(24.2)	105(25.8)	488(26.1)	
	Total	666(100.0)	453(100.0)	319(100.0)	387(100.0)	1825(100.0)	
Hyperlipidemia	No ^4^	573(86.2)	404(89.5)	265(85.8)	316(82.6)	1558(86.2)	0.0779
	Yes	93(13.8)	49(10.5)	54(14.2)	71(17.4)	267(13.8)	
	Total	666(100.0)	453(100.0)	319(100.0)	387(100.0)	1825(100.0)	

^1^ Q1: No milk intake, Q2: 0 < times/week ≤ 1, Q3: 1 < times/week ≤ 3, Q4: 3 > times/week. ^2^
*p* value by chi-square test. ^3^ No: not current smoker; Yes: current smoker ^4^ No: disease not diagnosed by doctor; Yes: disease diagnosed by doctor.

**Table 5 nutrients-14-01092-t005:** Framingham Risk Score (FRS) according to cow’s milk intake frequency.

Variables ^3^	Standard Score Range	Q1 ^1^ (n = 666)	Q2 (n = 453)	Q3 (n = 319)	Q4 (n = 387)	Total (n = 1825)	*p* Value ^2^
Age (years)	−9–8	7.0 ± 0.1 ^3^	7.0 ± 0.1	6.9 ± 0.1	7.0 ± 0.1	7.0 ± 0.1	0.6904
Total cholesterol	−2–3	0.3 ± 0.0	0.3 ± 0.1	0.3 ± 0.1	0.3 ± 0.1	0.3 ± 0.1	0.9950
HDL-cholesterol	−3–5	0.1 ± 0.1 ^a4^	−0.1 ± 0.1 ^a^	−0.1 ± 0.1 ^a^	−0.4 ± 0.1 ^b^	−0.1 ± 0.1	0.0184 *
Systolic blood pressure	−1~7	1.1 ± 0.1	0.9 ± 0.1	0.9 ± 0.1	0.9 ± 0.1	1.0 ± 0.1	0.4534
Diabetes status	0~4	0.3 ± 0.0	0.4 ± 0.1	0.4 ± 0.1	0.3 ± 0.1	0.3 ± 0	0.3818
Smoking status	0~2	0.1 ± 0.0	0.1 ± 0.0	0.1 ± 0.0	0.1 ± 0.0	0.1 ± 0	0.7980
Total FRS score	−3~37	8.9 ± 0.2 ^a^	8.5 ± 0.2 ^ab^	8.4 ± 0.3 ^ab^	8.3 ± 0.2 ^b^	8.6 ± 0.1	0.0277 *
10-year coronary heart disease (CHD) risk (%)	1~>30	9.4 ± 0.3 ^a^	8.9 ± 0.3 ^ab^	8.6 ± 0.3 ^ab^	8.5 ± 0.3 ^b^	8.9 ± 0.2	0.0490 *
AI ^5^		3.06 ± 0.04 ^a^	2.94 ± 0.06 ^ab^	2.89±0.06 ^b^	2.83±0.06 ^b^	2.95 ± 0.03	0.0060 **
AIP ^6^		0.37 ± 0.01 ^a^	0.31 ± 0.01 ^b^	0.31±0.02 ^b^	0.32±0.02 ^b^	0.33 ± 0.01	0.0032 **

^1^ Q1: No milk intake, Q2: 0 < times/week ≤ 1, Q3: 1 < times/week ≤ 3, Q4: 3 > times/week. ^2^
*p* value by proc survey multiple regression adjusted for age, energy intake, and household income (* *p* < 0.05, ** *p* < 0.01). ^3^ Mean ± SE. ^4^ ab: Values with different letters in the same row are significantly different at *p* = 0.05 by Bonferroni test. ^5^ AI (atherogenic index) = (blood non-HDL-cholesterol)/blood HDL-cholesterol. ^6^ AIP (atherogenic index of plasma) = log (blood TG/blood HDL-cholesterol).

**Table 6 nutrients-14-01092-t006:** Correlations among milk intake, calcium intake, and indicators of CHD risk.

	Milk Intake	Calcium Intake	FRS	AI	AIP
Milk intake	1.000 ^1^	0.626 ***	−0.048 *	−0.089 **	−0.040
Calcium intake	0.626 ***	1.000	−0.057 **	−0.053 *	−0.033
FRS	−0.048 *	−0.057 **	1.000	0.575 ***	0.540 ***
AI	−0.089 ***	−0.053 **	0.575 ***	1.000	0.715 ***
AIP	−0.040	−0.033	0.540 ***	0.715 ***	1.000

FRS: Framingham Risk Score. AI: atherogenic index. AIP: atherogenic index of plasma. ^1^ Pearson’s partial correlation coefficient (r), adjusted for age, energy intake, and household income. * *p* < 0.05, ** *p* < 0.01, *** *p* < 0.001: Significance as determines by Pearson’s correlation coefficient.

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
