# Peer review of "Cow’s Milk Intake and Risk of Coronary Heart Disease in Korean Postmenopausal Women"

_nutrients, 2022, doi:10.3390/nu14051092_

Round 1

Reviewer 1 Report

 Article

Milk intake and risk of coronary heart disease in Korean post-menopausal women

The manuscript aimed to identify the relationship between milk intake and coronary hearth disease risk in postmenopausal women using data from the 6th period of the Korea National Health and Nutrition Examination Survey (2013-2015).

Although the topic is of great interest, major revisions are required in order to improve the manuscript:

  • In the abstract, “better nutritional status” is reported in line 31. However, some information about nutritional status are lacking (i.e. detailed body composition evaluation as fat mass, blood parameters as glycemia…). Maybe, it would be better to either change the sentence as “…women who consume milk frequently had higher HDL levels, and lower level of CDH risk…” or adding information about evaluated parameters related to nutritional status (i.e. body weight, BMI…).
  • In the introduction, it could be useful to add information about the Dietary Reference Intake for Koreans (KDRIs). This would be helpful for international readers in order to better understand comparisons and percentage calculations. It would also be important to add information about milk reference intakes and, if possible, details about milk’s nutrients coverage of daily nutrients intake (i.e. how many energy/proteins/carbohydrates should come from milk every day).
  • There are some aspects in “Materials and Methods” section that should be better explained in order to understand obtained results:
    • It is important to describe milk characteristics above all in terms of fat contents. It is nutritionally different to talk about full-fat milk, semi-skimmed or skimmed milk consumption. Also in the introduction, it is underlined the role of milk fat. Thus, it is important to better explain this part. This is important also when table 3 is considered. For example, energy covered by milk consumption is strictly dependent on milk characteristics in term of fat content.
    • This study talks about milk consumption without taking into account the other aspect of the diet, maybe this should be better explained (diet was not controlled so this could be a confounding factor).
    • Lines 117-118, explain why those nutrients were chosen. For example, why iron?
    • It is difficult to understand FRS scoring calculation. In particular, the column “Standard score range” in table 6 is difficult to be understand by reading the paragraph 2.5 of “Materials and Methods” (line 134). For example, HDL-cholesterol range (-3 to 5 points) is -2 to 2 in table. Can you explain how to calculate this score?
  • In table 2, BMI classification reported in table’s note is different from the WHO classification. The reference should be reported.
  • In table 3, it is would be better to add information about KDRI standards, as said before. This could be explained in “introduction” section and add in table as note to make it better understandable.
  • Table 4 reported BMI values that are strange, maybe there was a mistake during table preparation.
  • Table 5, can you explain how did you calculate chi-squared test and why have you chosen that test?
  • Lines 272-280 should be rewrite by explaining why FFQ was chosen instead of 24-recall. The two sentences should be linked together underlining the differences obtained by choosing either FFQ or 24recall.
  • In line 287 both milk and calcium intake were reported. It should be important to consider also the diet and other possible calcium-rich food consumed. Maybe, it could be useful to describe subjects’ food habits in order to have an idea about the context and about other possible calcium sources (i.e. water typology, dairy products…).

Few minor revisions are required:

  • When talking about milk intake, it would be better to report it in mL instead of g
  • Line 13, and through the manuscript, 6th should be written as 6th
  • Line 23 “the” should be deleted
  • In line 196 “vitamin c” should be written as “vitamin C”.
  • The discussion can be improved by adding other studies regarding other possible works conducted all over the worlds about this topic. In post-menopausal women there is a lack of study but something about general population can be added to describe the situation.

Author Response

Thank you for reviewing the manuscript and for your meaningful comments. The things you pointed out have been corrected as much as possible. Thank you again. Revisions and answers for each question pointed out are as follows.

Reviewer 1

The manuscript aimed to identify the relationship between milk intake and coronary heart disease risk in postmenopausal women using data from the 6th period of the Korea National Health and Nutrition Examination Survey (2013-2015).

Although the topic is of great interest, major revisions are required in order to improve the manuscript:

In the abstract, “better nutritional status” is reported in line 31. However, some information about nutritional status are lacking (i.e. detailed body composition evaluation as fat mass, blood parameters as glycemia…). Maybe, it would be better to either change the sentence as “…women who consume milk frequently had higher HDL levels, and lower level of CDH risk…” or adding information about evaluated parameters related to nutritional status (i.e. body weight, BMI…).

ANSWER:

Line 31_revised: postmenopausal women who consumed milk frequently had the better nutritional status of calcium, phosphorus, and vitamin B12

In the introduction, it could be useful to add information about the Dietary Reference Intake for Koreans (KDRIs). This would be helpful for international readers in order to better understand comparisons and percentage calculations. It would also be important to add information about milk reference intakes and, if possible, details about milk’s nutrients coverage of daily nutrients intake (i.e. how many energy/proteins/carbohydrates should come from milk every day).

ANSWER: The following have been added to the introduction.;

Dietary Reference Intake for Koreans (KDRIs, Ministry of Health and Welfare, The Korean Nutrition Society. Dietary reference intakes for Koreans 2020. Sejong; 2020.) are energy and nutrient intake standards set for maintaining and promoting the health of the Korean people. To evaluate the adequacy of nutritional status, the subject's daily nutrient intake is gener-ally calculated as % KDRIs. According to the 2020 KDRIs, the recom-mended intake of calcium for women aged 50-64 years is 800 mg/day. Calcium is  the most deficient nutrient for Koreans of all ages [15].  Therefore, the Korean Nutrition Soci-ety recommends drinking one cup (200 ml) of cow’s milk per day for adults and two cups (400 mL) or more for growing children or pregnant women.  One cup (200 mL) of cow’s milk contains approximately 200 mg, which is 28.6% of the KRDI.

There are some aspects in “Materials and Methods” section that should be better explained in order to understand obtained results:

ANSWER:  Matched the order of the research method and the order of the results

2.3. General Characteristics of the Subjects

2.4 Evaluation of Nutrient Intakes Compared with the KDRIs

2.5 Anthropometric Measurements and Blood Profiles

2.6 Smoking Status and Disease Fraction (%)

2.7. Framingham Risk Score (FRS), Atherogenic index (AI), and Atherogenic index of plasma (AIP)

2.8. Statistical Analysis

It is important to describe milk characteristics above all in terms of fat contents. It is nutritionally different to talk about full-fat milk, semi-skimmed or skimmed milk consumption. Also in the introduction, it is underlined the role of milk fat. Thus, it is important to better explain this part. This is important also when table 3 is considered. For example, energy covered by milk consumption is strictly dependent on milk characteristics in term of fat content.

ANSWER: Added the following to the introduction & discussion part

Introduction:

The types of cow’s milk (non-fat/low-fat/ whole milk) or types of dairy products (milk, yogurt, cheese) consumed in adults, not postmenopausal women, were associated differentially with CHD. Because different types of dairy products (milk, cheese, and yogurt) have different nutritional content, recent studies have focused on the effects of CHD on the specific type of dairy products rather than total dairy intake [12-14]. Some studies reported a positive association of high-fat milk and CHD risk and an inverse association of low-fat milk/non-fat with CHD risk [7-9]. Opposite studies revealed that higher milk intake, regardless of the fat content of milk, was associated with a lower risk of CHD [10-12, 39] or neutral association [38,13,14].

Discussion:

In this study, among the milk drinkers, the proportion of drank regular milk, low-fat milk, and both-milk appeared 45.7%, 27.4%, and 10.1% respectively (data not shown), and the proportion among milk groups(Q2-Q4) was similar. In addition, the number of sub-jects in this study was not sufficient to classify the subjects according to milk type. Thus, the milk intaker in this study was defined as those who consumed any type of cow’s milk (regular cow or low-fat milk). In the case of Korea, milk intake is still insufficient in all ag-es [24,25,28], and thus the risk of CHD caused by the excessive consumption of whole milk is considered lower than that of Western countries. However, there still exists the possibility that excessive consumption of high-fat milk may increase the risk of CHD. A future study will examine intake of low-fat and high-fat milk and CHD risk.

This study talks about milk consumption without taking into account the other aspect of the diet, maybe this should be better explained (diet was not controlled so this could be a confounding factor).

ANSWER: Added the end of the discussion part

In this study, it is unclear if the milk consumed by the study subjects was low-fat or regular milk. Diet (high-fat foods, fruits, vegetables, and whole grains) could be an influencing factor for CHD risk, but it was not considered in this study. Those are other limitations of this study.

Lines 117-118, explain why those nutrients were chosen. For example, why iron?

Answer :

Important nutrients for adults as well as nutrients provided by KNHENS were selected. In case of Iron, postmenopausal women have a high iron concentration in their body because of menopause which could act as an oxidant. A study showed that total iron intake and serum iron concentrations were inversely associated with CHD incidence, but heme iron intake was positively related to CHD incidence.

It is difficult to understand FRS scoring calculation. In particular, the column “Standard score range” in table 6 is difficult to be understand by reading the paragraph 2.5 of “Materials and Methods” (line 134). For example, HDL-cholesterol range (-3 to 5 points) is -2 to 2 in table. Can you explain how to calculate this score?

Answer :

In the method, it describes the score range (FRS score range) for all age groups and genders (30-74 years old men and women), and Result 3 shows the score range of 50-64 years old women who are the subjects of this study. As pointed out, it may cause confusion, so the result table has been revised (same as the research method).

In table 2, BMI classification reported in table’s note is different from the WHO classification. The reference should be reported.

Answer: Revised as the bellows.

Obesity was classified into three categories according to body mass index (BMI, weight(kg)/ height(m) 2) values: less than 18.5 kg/m2 (underweight), less than 18.5-24.9 kg/m2 (normal weight), and over 25 kg/m2 (obesity).

In table 3, it is would be better to add information about KDRI standards, as said before. This could be explained in “introduction” section and add in table as note to make it better understandable.

Answer: Added the introduction and method part;

In a method-added those below sentences.

The KDRIs included 4 levels (estimated average requirement (EAR), recommended nutrient intake (RNI), adequate intake (AI), and tolerable upper intake Level (UL). To evaluate their adequacy, the energy and nutrient intakes for 11 nutrients (energy, pro-tein, calcium, phosphorus, iron, sodium, vitamin A, thiamine, riboflavin, niacin, and vitamin C) were compared with the age- and gender-specific values of the KDRIs. For energy, the intake ratio to the estimated energy requirement (EER) was calculated, and for nutrients, the intake ratio to the RNI or AI of each nutrient was calculated.

Table 4 reported BMI values that are strange, maybe there was a mistake during table preparation.

Answer: corrected BMI values under the table 4

 less than 18.5 kg/m2 (underweight), less than 18.5-24.9 kg/m2 (normal weight), and over 25 kg/m2 (obesity).

Table 5, can you explain how did you calculate chisquared test and why have you chosen that test?

Answer:

Diseases in Table 4 were coded as 0 and 1, respectively, and milk intake variables were coded into 4 groups (1,2,3,4), so the significance between each disease presence and milk group was chi-tested, respectively. We also edited the title of table 4 because the term “prevalence” in the table could cause confusion for readers.

.Lines 272-280 should be rewrite by explaining why FFQ was chosen instead of 24-recall. The two sentences should be linked together underlining the differences obtained by choosing either FFQ or 24recall.

Answer: Revised as the bellows.

Dietary survey methods include the 24-hour recall method and the FFQ method [21] The 24-hour recall method has a drawback in that the investigation is limited to one day. Semi-quantitative FFQ allows an investigation of general eating habits through a single survey [21]. However, studies reported that large-scale studies based on the FFQ underestimated energy or some nutrients because FFQ utilized a fixed food list [21, 21,22]. Ap-proximately 80% of postmenopausal women did not consume milk on the day of the study using 24-recall data. Only 36% of the subjects in FFQ data analysis did not drink milk. Previous Korean studies [23,24] also reported similar results to this study. Therefore, in this study, using the weekly cow’s milk intake frequency of FFQ, subjects were recruited, but the energy and nutrient intakes were determined from the 24-hour recall data of the subjects because of the FFQ disadvantages mentioned above.

In line 287 both milk and calcium intake were reported. It should be important to consider also the diet and other possible calcium-rich food consumed. Maybe, it could be useful to describe subjects’ food habits in order to have an idea about the context and about other possible calcium sources (i.e. water typology, dairy products…).

Answer:

Thanks for the good comment. Unfortunately, we did not investigate those things in this study. So, we added that as a limitation in the discussion section. However, we will include them in the next study.

Few minor revisions are required:

When talking about milk intake, it would be better to report it in mL instead of g

--- corrected

Line 13, and through the manuscript, 6th should be written as 6

--- corrected

Line 23 “the” should be deleted

--- corrected

In line 196 “vitamin c” should be written as “vitamin C”.

--- corrected

The discussion can be improved by adding other studies regarding other possible works conducted all over the worlds about this topic. In post-menopausal women there is a lack of study but something about general population can be added to describe the situation.

Answer: Added those below contents in discussion parts

The types of cow’s milk (non-fat/low-fat/ whole milk) or types of dairy products (milk, yogurt, cheese) consumed in adults, not postmenopausal women, were associated differ-entially with CHD. Because different types of dairy products (milk, cheese, and yogurt) have different nutritional content, recent studies have focused on the effects of CHD on the specific type of dairy products rather than total dairy intake [12-14]. Some studies reported a positive association of high-fat milk and CHD risk and an inverse association of low-fat milk/non-fat with CHD risk [7-9]. Opposite studies revealed that higher milk intake, re-gardless of the fat content of milk, was associated with a lower risk of CHD [10-12, 39] or neutral association [38,13,14].

In this study, among the milk drinkers, the proportion of drank regular milk, low-fat milk, and both-milk appeared 45.7%, 27.4%, and 10.1% respectively (data not shown), and the proportion among milk groups(Q2-Q4) was similar. In addition, the number of sub-jects in this study was not sufficient to classify the subjects according to milk type. Thus, the milk intaker in this study was defined as those who consumed any type of cow’s milk (regular cow or low-fat milk). In the case of Korea, milk intake is still insufficient in all ages [24,25,28], and thus the risk of CHD caused by the excessive consumption of whole milk is considered lower than that of Western countries. However, there still exists the possibility that excessive consumption of high-fat milk may increase the risk of CHD. A future study will examine intake of low-fat and high-fat milk and CHD risk

Thank you again for your efforts and meaningful comments.

Reviewer 2 Report

This manuscript providing detailed cross-sectional analyses from the 2013-2015 KNHANES survey was read with interest. Questions, comments, and suggestions are noted below for the author’s consideration.

Specific Comments/Suggestions:

Page 2, Line 81. Could it please be clarified how “insincere answers” were determined and if there were specific variables these related to? For example, was this based on responses being considered outliers/more than 2 standard deviations away from the mean or median for specific variables?

Page 4, Lines 160-161 & Page 8, Line 241: Why were age, energy and house income chosen as potential confounders? Was a stepwise analysis conducted to determine that these three variables were the most appropriate to use as possible confounders? Was a crude or other models of confounders considered utilized? Later in the discussion the justification for the use of these variables as possible confounders was described; however, this explanation would be useful in the methods section for context.

Is data available for the types of milk, such as the fat content/flavoured milk varieties available? If so, could this be mentioned in the methods and/or results to indicate the possible types of milk available along with their proportion of intake?

Methods, Page 3, lines 114 to 122. Why was the calcium intake determined based on 24-hour recall as opposed to FFQ, even though it was noted in the discussion that the FFQ w(and other nutrient) assessment based on the 24-hour recall method validated? Also, what food sources were considered? Was supplement use also considered? Were any potential inhibitory nutrients (such as phytates) accounted for in the analyses?

For Table 3, lines 197 to 198; Table 4, lines 212 to 213; & Table 6, lines 237 to 238. Please replace the word “alphabets” with “letters”, and the word “raw” to “row” to correct the sentence. Also, were these comparisons between quantiles also conducted for the content in Tables 2 and 5 and no differences were observed or are differences observed here as well and not noted?

Statistical analyses: Page 4, Lines 163 to 164. On this page, it is noted that the correlation between milk intake frequency and variables was analyzed using the Spearman's correlation coefficient. However, on Page 8, Lines 241 to 243, it is stated that “Pearson correlation” was utilized in the statement, “After adjusting for energy, age, and house income, the correlations between milk frequency (times/week), milk intake (g/d), calcium intake (g/d), FRS, AI, and AIP were analyzed by Pearson correlation (Table 7).” Please clarify.

Page 8, Lines 243 to 245: For the statement, “The frequency of milk intake was significantly and negatively correlated with the CHD risk-indicators FRS, AI, and AIP, and significantly and positively correlated with daily milk intake (g/d).” Given the amount of daily milk intake was based on the frequency data presented (Page 3, lines 98 to 101) this seems as though these two variables would have collinearity given the amount of daily milk intake is calculated based on the frequency of milk intake and hence these two variables would be correlated. Could the interpretation and reason for both analyses please be justified?

Discussion, Page 8, Lines 263 to 265. It is noted that “the two main issues addressed in this study are measurement of milk intake in postmenopausal women measured by FFQ,…” however, the present study does not appear to be a validation study of the use of the FFQ in this population, instead of an issue being addressed, are the authors trying to say that the present analyses provides a summary of milk intake in the noted population? Please clarify.

Discussion, Page 9, Lines 310 to 312. Modify to ensure it is consistently TC being referred to as opposed to the current TC and TG mentioned.

Thank-you for your time and consideration of these suggestions.

Reviewer 3 Report

line 40: change from during menopause to with menopause

line 42: add years after 50

lines 48-50 can be edited simply to Milk is a nutrient dense food.

lines 50-61 would benefit from editing to improve flow

line 67: should reference #16 be included, currently 17-20? 

lines 85 & 86: was there analysis to compare KNHANES participants who were and were not included in the study for any demographic characteristics? If not, provide rationale.

line 95: add a week after 5-6 times

line 101: 1 cup is equal to 240 mL. Was 200mL used for the daily intake of milk calculation? If so, provide rationale.

2.2 Describe the methodological approach for determining quartiles in text.

Remove Table 1

2.6 Define BMI categories after describing how BMI is calculated (ln 126). Categories are used to for data analysis in addition to BMI value.

2.5 I assume the FRS is for 10-yr CHD risk. Authors should be explicit about which FRS index was used in the study. I reached my assumption by reviewing the data utilized and closely looking at the indices, but believe authors could improve clarity for readers quite easily. 

2.6 Authors are encouraged to clarity which statistical test is used to compare means across quartiles. Posthoc test is identified so that is good.

ln 164: authors state "Spearman's" (should be Spearman) correlation OR is this an error? Table 7 reports Pearson's correlation

Table 2: this table needs some revision

Editing recommended to remove some information from the footer (mean ± SE and frequency (%)) to the title or within the table for improved presentation of data to readers.

Change variable title from House income to Household income

Change Obesity to BMI Category and change footer to reflect use of categories and addition to Methods section 

Footer: correct direction </> for underweight, BMI>18.5 to underweight BMI<18.5

3.3 suggest removing correlations table and including results in the text in this section

ln 207: suggest change "with no" to "failed to reach" statistical significance.

Remove table 5 and report results in text.

3.5 consider adding variance (SE) to average when presenting data in text although the variance is available in Table 6

3.6 remove table 7 and present results in text, perhaps section 3.3 (clarification needed whether Pearson's correlation or Spearman correlation were used related to discrepancy between Materials and Methods and Results sections)

ln 336: change "proper" milk intake to "increased" milk intake is related to a more favorable CHD risk profile.

lns 358-359: conclusion overstates findings and are misleading; milk intake for Q4 averaged 215.9±4.6 g/d - this is less than 1 cup of milk

ln 362  and 364 conclusion overstates findings and are misleading; suggest perhaps additional research exploring milk as an intervention for postmenopausal women as a means to reduce CHD risk are needed.

General comments:

I assume this is cow's milk (dairy). Author's could improve clarity by addressing this in the introduction and methods sections. There are many types of milk available to consumers.

Author Response

Reviewer 3

Thank you for reviewing the manuscript and for your meaningful comments. The things you pointed out have been corrected as much as possible. Thank you again. Revisions and answers for each question pointed out are as follows.

Comments and Suggestions for Authors

line 40: change from during menopause to with menopause

Answer:

-Revised 

line 42: add years after 50

Answer:

-Revised 

lines 48-50 can be edited simply to Milk is a nutrient dense food.

Answer:

-Revised

lines 50-61 would benefit from editing to improve flow

Answer:

Revised as the below:

The traditional risk factors for CHD include dyslipidemia, hypertension, diabetes, obesity, family history, and smoking, most of which can be controlled through individual lifestyles [3]. To prevent CHD, it is recommended to increase the intake of vegetables, fruit, and complex carbohydrates and reduce the intake of foods containing saturated fat and cholesterol [6]. Although milk is a nutrient-dense food, there have been conflicting results of milk/ dairy products intake and CHD [7-14]. The effects of milk and dairy products on the risk of CHD differ according to the type of dairy products consumed (milk, yogurt, cheese) or milk fat content (low-fat milk /regular milk) [7-14]. Some studies have reported that excessive consumption of milk and dairy products increased the risk of CHD because of high saturated fat content [7-9]. Positive effects on milk and dairy products and CHD have also been reported; sufficient intake of milk and dairy products lowered blood pressure and increased HDL-cholesterol levels [10-11].

line 67: should reference #16 be included, currently 17-20?

Answer:

-Revised 

lines 85 & 86: was there analysis to compare KNHANES participants who were and were not included in the study for any demographic characteristics? If not, provide rationale.

Answer:

In this study, only postmenopausal women aged 50-64 years were included. Since many studies on adults and men of the same age already exist, this study only focused on postmenopausal women aged 50-64 years. (In the case of women aged 50-64, almost all women of the age group were postmenopausal women, and non-menopausal women accounted for about 1%, so they were excluded from the study). However, since your suggestion is meaningful, we will analyze it further and use it in the next thesis.

line 95: add a week after 5-6 times

Answer:

-Revised

line 101: 1 cup is equal to 240 mL. Was 200mL used for the daily intake of milk calculation? If so, provide rationale.

Answer:

In Korea, the amount of one cup is set at 200 mL by the Korean Nutrition Society, so the FFQ of KNHANES (per one serving = 200 mL). In the case of Koreans, because their physique is smaller than that of Westerners, the serving size of food for Koreans was set to smaller than that of Westerners.

2.2 Describe the methodological approach for determining quartiles in text.

Answer:

In this study, subjects were first classified as non-milk drinkers and drinkers, and then milk drinkers were divided into thirds by the proc rank method.

The subject classification method is presented in Section 2.3.:

Based on the frequency of milk intake per week, total subjects were classified into 4 groups: Q1, group that did not drink milk (no milk, n=666); Q2, 0< frequency of milk intake per week <1 (n=453); Q3, 1< frequency of milk intake per week <3 (n=319); and Q4, frequency of milk intake >3 times per week (n=387)

Remove Table 1

Answer:

As you suggested, we erased table 1

2.6 Define BMI categories after describing how BMI is calculated (ln 126). Categories are used to for data analysis in addition to BMI value.

Answer:

The range of obesity was also modified in the study method and tables’ footnote.

Obesity was classified into three categories according to body mass index (BMI, weight(kg)/ height(m) 2) values: less than 18.5 kg/m2 (underweight), less than 18.5-24.9 kg/m2 (normal weight), and over 25 kg/m2 (obesity).

2.5 I assume the FRS is for 10-yr CHD risk. Authors should be explicit about which FRS index was used in the study. I reached my assumption by reviewing the data utilized and closely looking at the indices, but believe authors could improve clarity for readers quite easily.

Answer:

As you said, it would have been better to present FRS components in a table, but we explained the FRS calculations with FRS components relatively detail in the research method, and the FRS score range was explained in Table 6. We would be grateful if you could understand my situation.

2.6 Authors are encouraged to clarity which statistical test is used to compare means across quartiles. Posthoc test is identified so that is good.

Answer:

As you suggested, we added the below in the statistical methods part.

Continuous variables are expressed as the mean and standard error, and significance between milk group and each variable was verified by proc survey regression after adjusting for age, house income, and energy intake. To verify the differences between groups, Bonferroni's post hoc test was performed at the level of α=0.05

ln 164: authors state "Spearman's" (should be Spearman) correlation OR is this an error? Table 7 reports Pearson's correlation

Answer:

Corrected –   Correlation between milk intake frequency and variables was analyzed using the Pearson’ partial correlation (control: age, energy intake, house income.

Table 2: this table needs some revision

Editing recommended to remove some information from the footer (mean ± SE and frequency (%)) to the title or within the table for improved presentation of data to readers.

Answer:

-Revised table 2

Change variable title from House income to Household income

Answer:

-Revised household in table 2

Change Obesity to BMI Category and change footer to reflect use of categories and addition to Methods section

Footer: correct direction </> for underweight, BMI>18.5 to underweight BMI<18.5

Answer:

It was modified as in the study method.; Obesity, divided by body mass index (BMI, kg/m2): underweight, BMI <18.5; normal weight, 18.5≤ BMI ≤24.9, obese, BMI ≥25.0.

3.3 suggest removing the correlations table and including results in the text in this section

Answer:

We revised the correlation table more clearly, and thus we want the table to be left alone. I would be grateful if you could understand.

ln 207: suggest change "with no" to "failed to reach" statistical significance.

Answer:

Revised : Blood triglyceride concentrations were lower in the Q2∽Q4 group than in the Q1 group, but failed to reach statistical significance

Remove table 5 and report results in text.

Answer:

As pointed out, in Table 5, since there is no statistical significance in the proportion of disease presence between the 4 groups, it is reasonable to delete the table and describe the results., hypertension). However, diseases (diabetes, hyperlipidemia, hypertension) are major factors of CHD. As seen in Table 5, there was no difference in this disease fraction between the 4 groups, which suggested that the low CHD risk (%) in the Q4 group is more likely related to high milk intake, not other factors such as diseases. So if that's okay with you, I'd like to leave it that way.

3.5 consider adding variance (SE) to average when presenting data in text although the variance is available in Table 6

Answer:

Revised all values in table 6 as Mean±SE

3.6 remove table 7 and present results in text, perhaps section 3.3 (clarification needed whether Pearson's correlation or Spearman correlation were used related to discrepancy between Materials and Methods and Results sections)

Answer:

The correlation was analyzed by Pearson's partial correlation after adjusting for age, energy intake, and household income.  Therefore, both the method and result table 7 were modified in the same context.

ln 336: change "proper" milk intake to "increased" milk intake is related to a more favorable CHD risk profile.

Answer:

Revised to increased milk intake

lns 358-359: conclusion overstates findings and are misleading; milk intake for Q4 averaged 215.9±4.6 g/d - this is less than 1 cup of milk

Answer:

Revised - The Korean postmenopausal women (50-64 years old) who consumed 4 times or more per week

ln 362  and 364 conclusion overstates findings and are misleading; suggest perhaps additional research exploring milk as an intervention for postmenopausal women as a means to reduce CHD risk are needed.

Answer:

Revised: The Korean postmenopausal women (50-64 years old) who consumed cow’s milk 4 times or more per week had higher intakes of calcium, phosphorus, and riboflavin than the group that did not consume milk, and also had significantly higher HDL-cholesterol in blood. In addition, the levels of CHD risk indicators such as FRS and AI were significantly lower in the high cow’s milk intake group. These findings suggest that further studies exploring cow’s milk as an intervention for postmenopausal women as a means to reduce CHD risk may be needed.

General comments:

I assume this is cow's milk (dairy). Author's could improve clarity by addressing this in the introduction and methods sections. There are many types of milk available to consumers.

Answer:

Revised milk as cow’s milk or milk and dairy products when necessary.

the milk group in this study was defined as those who consumed any type of cow’s milk (regular cow or low-fat milk).

Thank you again for your efforts and meaningful comments.
